# Evaluation of Genomic Typing Methods in the *Salmonella* Reference Laboratory in Public Health, England, 2012–2020

**DOI:** 10.3390/pathogens12020223

**Published:** 2023-01-31

**Authors:** Marie Anne Chattaway, Anaïs Painset, Gauri Godbole, Saheer Gharbia, Claire Jenkins

**Affiliations:** 1Gastrointestinal Bacteria Refence Unit, United Kingdom Health Security Agency (UKHSA), London NW9 5HT, UK; 2Gastrointestinal Pathogens and Food Safety (One Health), United Kingdom Health Security Agency (UKHSA), London NW9 5HT, UK

**Keywords:** *Salmonella*, genomics, molecular typing, England, UKAS

## Abstract

We aim to provide an evidence-based evaluation of whole genome sequence (WGS) methods, employed at the *Salmonella* reference laboratory in England, in terms of its impact on public health and whether these methods remain a fit for purpose test under UKAS ISO 15189. The evaluation of the genomic methods were mapped against the value of detecting microbiological clusters to support the investigation of food-borne outbreaks of *Salmonella* in England between 2012–2020. The analysis of WGS with both SNP- and allelic-based methods provided an unprecedented level of strain discrimination and detection of additional clusters when comparing to all of the previous typing methods. The robustness of the routine genomic sequencing at the reference laboratory ensured confidence in the microbiological identifications, even in large outbreaks with complex international food distribution networks. There was evidence that the phylogeny derived from the WGS data can be used to inform the provenance of strains and support discrimination between domestic and non-domestic transmission events. Further insight on the evolutionary context of the emerging pathogenic strains was enabled with a deep dive of the phylogenetic data, including the detection of nested clusters. The public availability of the WGS data linked to the clinical, epidemiological and environmental context of the sequenced strains has improved the trace-back investigations during outbreaks. The global expansion in the use of WGS-based typing in reference laboratories has shown that the WGS methods are a fit for purpose test in public health as it has ensured the rapid implementation of interventions to protect public health, informed risk assessment and has facilitated the management of national and international food-borne outbreaks of *Salmonella*.

## 1. Background

*Salmonella* is a gram negative rod belonging to the family Enterobacteriales, and human infection is primarily caused by the *Salmonella enterica* subspecies *enterica*. Clinically speaking, S. enterica is further subdivided into the invasive typhoidal group, which causes enteric fever (S. Typhi, S. Paratyphi A, B and C), often requiring treatment and can lead to further complications if not treated [1,2,3]. Enteric fever strains can cause household outbreaks, but are predominantly associated with endemic countries, with importation into the UK due to travel [4]. The other clinically relevant group is non-typhoidal *Salmonella* (NTS); although they generally cause a self-limiting gastroenteritis, they can cause invasive disease, such as bacteremia [5], lesions [6] or osteomyelitis [1]. The reporting of *Salmonella* isolated from human clinical diagnostic samples in public health laboratories is mandatory under national legislation, and the United Kingdom Health Security Agency (UKHSA), Gastrointestinal Bacteria Reference Unit (GBRU) receives approximately 10,000 presumptive *Salmonella* isolates each year from diagnostic microbiology laboratories, private laboratories, industry and food, water and environmental (FW&E) laboratories for confirmation of the identity and typing [7,8]. NTS pose an important public health threat with zoonotic infections, food safety and the containment of foodborne outbreaks, which have resulted in higher hospitalisation rates when the demographics are associated with young children. It is therefore essential that *Salmonella* are prospectively typed to detect any emerging threats and to put public control measures into place.

Historically, the species *Salmonella enterica* was classified into serovars on the basis of the extensive diversity of the lipopolysaccharide (O) antigens and the flagellar protein (H) antigens [9], based on the scheme developed by White [10] and extended by Kauffmann in the 1940s [11,12,13,14]. Today, over 2500 antigenically distinct serovars have been recognised [15]. Further differentiation using phage typing for the most clinically important serovars (e.g., typhoidal *Salmonellae*) and/or those that are most frequently detected (e.g., *S.* Typhimurium [16,17,18] and *S.* Enteritidis [19]) has been employed since the 1930s. Phage typing was rapid and cost effective, but lacked discrimination, and the phage type (PT) was not a robust marker of the genetic relatedness [20]. Therefore, molecular subtyping techniques were developed, including pulse-field gel electrophoresis (PFGE) [21,22,23] and a variable number of tandem repeats -based fingerprinting (VNTR), also known as multi-locus variable number of tandem repeats analysis (MLVA) [24,25,26,27]. Due to the laborious nature of these methods, in England, these were used reactively in outbreaks identified by epidemiological links.

Whole genome sequencing (WGS) involves the sequencing of the whole genome, in which the data can then be used in downstream analysis, such as strain identification and subtyping, to assess the strain relatedness. In addition, WGS can be upscaled and used to streamline the laboratory process, reduce processing and turnaround times, increase the fine typing discriminatory power for emerging outbreaks, improve food safety and bridge one-health surveillance [28].

In 2015, the Gastrointestinal Bacteria Reference Unit (GBRU) at the United Kingdom Health Security Agency (UKHSA, formally Public Health England) fully implemented WGS methods based on clonal inference via E-burst groups (eBG) [29] and Single Nucleotide Polymorphism (SNP) typing [30] as a replacement for the use of the conventional methods described above [8,28]. The United Kingdom Accreditation Service (UKAS) is the national accreditation body for the United Kingdom and are appointed by the government to assess and accredit organisations that provide services. GBRU are accredited to the International Organisation for Standardization (ISO) 15189:2012 standards (Number 8197) for medical laboratories, including the use of WGS methods. A requirement of the ISO 15189 standards is to ensure that tests are reviewed and continue to be fit for purpose. The aim of this perspective study was to review the evidence base for the utility of WGS for the identification and typing of *Salmonella* in the reference laboratory and the utilisation of these typing methods for the detection and investigation of outbreaks.

## 2. Systematic Search Methods

To provide an evidence base for using genomic microbiological typing methods for public health impact, a systematic review was performed using the ‘Preferred Reporting Items for Systematic Reviews and Meta-Analysis’ (or ‘PRISMA’) guidelines [31]. The relevant English articles available in Medline (Pubmed) were retrieved using the predefined search terms: ‘*Salmonella*’, ‘sequencing’, ‘outbreak’ and ‘England’ (Appendix A). The literature search was conducted until the end of February 2020. The eligibility of 17 published reports in this review was based primarily on whole genome sequencing (WGS), in which additional typing techniques may or may not have been used, including phage typing (PT), multi-locus variable number of tandem repeats analysis (MLVA) or pulse-field gel electrophoresis (PFGE). Articles were included for this review provided they reported the use of WGS (short read) in association with surveillance and/or outbreak investigations (Table 1).

In addition, a review of joint published *Salmonella* risk assessments between the European Centre for Disease Prevention and Control (ECDC), the European Food Safety Authority (ECDC) and the UKHSA was performed. Relevant risk assessments available via ECDC (Risk assessments (europa.eu) were retrieved until the end of February 2020. The eligibility of the published risk assessments in this review was based primarily on WGS in which additional typing techniques may or may not have been used (PT, MLVA or PFGE) and 24 risk assessments were included where cases were found in England (Appendix A).

## 3. Methods Applied for Sequencing and Typing

### 3.1. Quality Metrics of Sequence Data and Identification of Salmonella

Since 2014, all *Salmonella* isolates referred to the GBRU were extracted on the QiaSymphony (Qiagen, Germany), and sequenced using the Illumina HiSeq 2500 platform (Illumina Inc., San Diego, CA, USA) in rapid run mode (2 × 100 bp reads). The quality of the raw FASTQ files is evaluated using an in-house program, qa_and_trim, which determines the metric yield of the sample (where yields of data from an isolate are below 150 Mb, they are repeated) and trims the files with Trimmomatic [47] (using the parameters LEADING:30, TRAILING:30, SLIDINGWINDOW:10:20, and MINLEN:50). All of the subsequent analysis is carried out on the trimmed files. As previously described, the KmerID pipeline (https://github.com/phe-bioinformatics/kmerid (accessed on 7 December 2022)) is used to compare the sequenced reads with the published genomes in order to identify the bacterial species and *Salmonella* subspecies [8] and detect potential contaminated sequences. The quality of the sample is further evaluated by MLST using MOST (https://github.com/phe-bioinformatics/MOST (accessed on 7 December 2022)) [48] associated with the Achtman seven gene scheme [29]. Each sample is assigned a “traffic light” colour, depending on its coverage and mapping metrics: Green, Amber or Red, reflecting the level of confidence in the ST determination. A green flag is associated with a maximum percentage of non-consensus bases <15%, minimum consensus reads depth >2, percentage coverage for each loci = 100%, and that the ST determination has not failed; the amber flag reflects a maximum percentage of non-consensus bases >15% or minimum consensus reads depth is between 0 and 5 (inclusive); a red flag is applied if the percentage coverage for any locus is <100% or the ST determination has failed. *Salmonella* serovar determination is predicted based on the *Salmonella* eBURST group (eBG) or Sequence Type (ST) [29] and checked against a validated UKHSA database [8].

All of the FASTQ sequences are made publically available by routinely uploading the *Salmonella* sequence data to NCBI BioProject PRJNA248792 (https://www.ncbi.nlm.nih.gov/bioproject/?term=PRJNA248792 (accessed on 7 December 2022)) and Enterobase to enable other institutions to utilise the data [49]. The basic metadata is provided, including the Month/Year, Country, Isolation source (e.g., human, animal, food), serovar and ST. On 30 January 2023, 73,558 SRA experiments were available for analysis.

### 3.2. SNP Typing

High quality reads are mapped to a specific reference *Salmonella* strain depending on the eBG (Appendix A), using Burrows-Wheeler Aligner—Maximum Exact Matching (BWA MEM) [50]. The output from BWA is sorted and indexed to produce a binary alignment map (BAM) using Samtools [50]. The Genome Analysis Toolkit (GATK) is then used to create a variant call format (VCF) file from each of the BAMs, which is further parsed to extract only the single nucleotide polymorphism (SNP) positions of high quality (mapping quality (MQ) > 30, depth (DP) > 10, variant ratio > 0.9) [51,52].

For each isolate, every change in the genome sequence (a SNP) relative to a reference genome is recorded and can be used to quantify the genetic relatedness between isolates. This is undertaken by hierarchical single linkage clustering, which is performed on the pairwise SNP difference between all of the isolates at descending distance thresholds (Δ250, Δ100, Δ50, Δ25, Δ10, Δ5, Δ0). The result of the clustering is a SNP profile, or SNP address, that is used to describe the population structure based on the clonal group membership, as indicated by the number at each level of the seven-number SNP address [30]. The resultant SNP address provides an isolate-level nomenclature, where two isolates with the same SNP addresses have 0 SNP differences. Single linkage clustering is employed when clustering isolates together and assigning SNP addresses; this means that at a given SNP threshold, all isolates will be related to at least one other isolate by the number of SNPs for that threshold (i.e., at the 5-SNP threshold (t5), all isolates will be related to another isolate by a maximum of 5-SNPs). The SNP address is often supported with phylogenetic analysis to describe the evolutionary relationship between the isolates (Table 1). All of the processes, including the SNP analysis, are accredited by the UKAS to the ISO 15189 standards.

### 3.3. cgMLST

EnteroBase is an integrated software environment that supports the identification of global population structures within several bacterial genera, including *Salmonella*. EnteroBase performs daily scans of the SRA, via its Entrez APIs [53] for novel Illumina short-read sequences, for each of the bacterial genera that it supports. It uploads the new reads and assembles them into annotated draft genomes, which are published if they pass quality control [49]. The core genome MLST (cgMLST) scheme has been defined in EnteroBase, as a standard genotyping method for *Salmonella* comprising of 3002 loci for the improved discrimination of the genotype as compared to 7-locus MLST. Hierarchical Clustering (HierCC) supports the analyses of population structures based on cgMLST at multiple levels of resolution [49]. To identify the cut-off values in the stepwise cgMLST allelic distances. which would reliably resolve the natural populations, a matrix of pairwise allelic distances (excluding pairwise missing data) is first calculated for all of the existing pairs of cgSTs, and one matrix for the HierCC cluster numbers at each level of the allelic distance, that is, one matrix for HC0, HC1, HC2, …, HC3001. A genus-specific subset of the most reliable HierCC clusters is reported by EnteroBase. For *Salmonella*, 13 HierCC levels are reported, ranging between HC0 (indistinguishable except for missing data) and HC2850 [49] which enables institutions to compare the data and detect emerging clusters on a global scale.

## 4. Evaluation of WGS Methods

### 4.1. Definition and Routine Identification of Salmonella Lineages

Whole genome sequencing has laid the framework to define the clonality of *Salmonella* to replace phenotypic methods. The definition of a *Salmonella* ‘lineage’ changes depending on the context of the study, and has been described in many ways; for example, in terms of serovar [54,55], sequence types [29,56] or distinct lineages within specific serovars/STs such as Typhimurium or Enteritidis [57,58,59,60]. In terms of applying genomic methods in reference microbiology, the lineages are defined in multifaceted layers: first by the species or subspecies, next by the eBG and ST (serovar reported for backward compatibility), and finally by the SNP profile.

*Salmonella* species and subspecies lineages were traditionally defined by biochemical properties, which was subjective to variable interpretation and added potential disparity in the subspecies definitions, routine typing, and the ability to detect novel subspecies. A large-scale analysis of the WGS data from the routine sequencing of clinical isolates was employed to define and characterise the *Salmonella* subspecies’ population structure. This evaluation study clearly demonstrated that the *Salmonella* species and subspecies were genetically distinct and the proposed species and subspecies structure was sufficiently biologically robust for routine application [42,61], which has now been implemented [8]. The defined lineages are as follows: (i) *Salmonella enterica* and *Salmonella bongori* remain as separate species, (ii) *S. enterica* subspecies IIIa should be reclassified as the separate species, *S. arizonae,* (iii) *S. enterica* subspecies enterica (I), *S. enterica* subspecies salamae (II), *S. enterica* subspecies diaraizonae (IIIb), *S. enterica* subspecies houtenae (IV) and *S. enterica* subspecies indica (VI) remain as *S. enterica* subspecies, (iv) five novel subspecies (*S. enterica* subsp. *londinensis* (VII), *S. enterica* subsp. *brasiliensis* (VIII), *S. enterica* subsp. *hibernicus* (IX), *S. enterica* subsp. *essexiensis* (X), *S. enterica* subsp. *reptilium* (XI) are identified as additional *S. enterica* subspecies (Figure 1). The implementation of this methodology has enabled UKHSA to completely withdraw the biochemical testing methods and collaborate with organisations to detect and define novel species, such as *Salmonella englandensis* (Figure 1). Further work is in progress to officially recognise the updated genomic *Salmonella* nomenclature with the International Journal of Systematic and Evolutionary Microbiology (IJSEM).

The lineages are then further defined in terms of E-Burst groups (EBG) and sequence types (ST) with the parallel reporting of the predicted antigenic formula or serovar nomenclature to retain the historical naming conventions and maintain the backwards compatibility with the information linked to these serovars. Although the use of Multi Locus Sequence Typing (MLST) as a replacement for serotyping had been validated [29] and implemented at the UKHSA [8,28], these methods did not provide a genomic framework for the more complex *Salmonella* serovars. For example, when multiple serovars belonged to the same sequence type, or how to define novel *Salmonella* sequence types and serovars as they evolved. In order to address this issue, a large-scale validation study was undertaken to genomically define the serovars of *Salmonella* using Major Antigenic Clustering or ‘MAC typing’ [63]. In this study, we analysed 46,000 isolates of the *Salmonella enterica* subspecies *enterica* to define the clusters in two stages: Multi Locus Sequence Typing followed by antigen prediction to provide a MAC type and reported as ST-Serovar. This genomic framework of defining serovar lineages was able to assign 99.96% of the isolates to a MAC type and is the current method used in the reference laboratory, enabling the complete withdrawal of phenotype serological testing [63].

Finally, the strain lineages are defined by the differences in the core genome, SNP typing or cgMLST [49], as described in the methods above [30]. It is this fine typing level that is used for determining the strains’ relatedness and used to support outbreak investigations. The application of genomic microbiological typing has been reported in individual outbreak investigations in England (Table 1), but not as a holistic evaluation, as is described in this study.

### 4.2. Preliminary Evaluations Studies and Comparisons with PFGE, PT and MLVA

The first study in England to compare the PFGE with the WGS data was a prospective comparison of 24 isolates from a multi-country outbreak of *S.* Newport, associated with the consumption of watermelon, in 2012. The sequences of the outbreak isolates from humans and food differed from each other, by 0–1 SNPs apart, compared to several thousand SNPs from the nearest non-outbreak strain [32]. In 2014, a retrospective investigation of an outbreak of *S*. Typhimurium DT8, where an increase in this phage type associated with the consumption of mayonnaise made from raw eggs was reported in the summer of 2012, was performed [36]. Whole genome sequencing (WGS) was used to retrospectively investigate this outbreak with a view to assess the similarity between the suspect food and the human isolates and to characterise a known point source outbreak to assist in the development of algorithms for outbreak detection [36]. The sequence data showed that the outbreak-associated isolates, including the food isolates, formed a tightly clustered monophyletic group, with a maximum pairwise distance of three single nucleotide polymorphisms. These two outbreak investigations provided preliminary evidence of the usefulness of WGS in linking human clinical cases to each other and to the contaminated food vehicle that caused the outbreak [36].

By 2015, the routine sequencing of *Salmonella* at the UKHSA was fully implemented and was used prospectively to investigate a prolonged restaurant outbreak of *S*. Typhimurium, MLVA profile (3–14–9-0–0211) [33]. Despite multiple controls, the outbreak continued to resurface, and the environmental sampling was extended to the restaurant drains which identified structural faults with the draining system as the source. The genomic analysis found that all of the outbreak strains clustered within a nationally unique five SNP cluster, with isolates differing between 0–5 SNPs [33].

These initial investigations showcased the highly discriminatory nature of WGS and provided preliminary evidence that isolates that fall within a five SNP single linked cluster (SLC) are more likely to be part of the same temporally linked outbreak and demonstrated that we were able to detect the same clusters, as defined by PFGE, PT and MLVA [32,33,36].

### 4.3. Studies Comparing SNP and cgMLST Analysis

Preliminary evaluation studies have highlighted that the use of SNP typing is a robust method for detecting outbreaks, but a key international method that is used for detecting clusters is core genome multi locus sequencing typing (cgMLST) [49,64]. Two large validation studies were performed to compare the SNP and cgMLST methods in the context of outbreaks associated with large European outbreaks in order to ascertain the concordance of these methods. In 2018, Pearce et al. [42] applied cgMLST methods to a large European outbreak of *S*. Enteritidis, which had been extensively characterised using SNP-based approaches [34,35]. This study found that the cgMLST cluster was congruent with the original SNP-based analysis, the date of isolation and the epidemiological data and confirmed that the genetic diversity among the isolates predated the outbreak associated with eggs from a German producer and was likely present at the infection source [42]. They concluded that the cgMLST scheme was shown to be a standardised and scalable typing method, which allowed *Salmonella* outbreaks to be analysed and compared across laboratories and jurisdictions.

A further study in 2019 reanalysed the data from another large European multi-country *S.* Enteritidis outbreak associated with Polish eggs, but with various European institutes using different analysis workflows to identify the isolates potentially related to the outbreak [46]. The objective in this case was to compare the output of six of these different typing workflows (distance matrices of either SNP-based or allele-based workflows) in terms of cluster detection and concordance. Copian et al. found that hierarchical clustering with the six different allele- and SNP-based typing workflows generated clusters with similar compositions. Further investigations found that, even in the absence of coordinated typing procedures, the various workflows that were currently in use by six European public-health authorities could identify concordant clusters of genetically related *S.* Enteritidis isolates, thus providing public-health researchers with comparable tools for the detection of infectious-disease outbreaks [46]. Hence, both SNP and cgMLST are appropriate methods for cluster detection and are both used interchangeably for international communications, including ECDC rapid risk assessments (Appendix A). The flexibility of reference laboratories to analyse the genomic data using the same raw data but applying different methodologies has enabled the comparative analysis of international outbreaks on a global scale. This has resulted in other reference laboratories being able to set up their own bespoke pipelines within each country, whilst maintaining the ability to use other methods, and a vital approach for inter-laboratory comparison analysis (Appendix A).

## 5. Downstream Applications of WGS Methods

### 5.1. Preliminary Evaluations Studies for Detection of Clusters with Respect to Epidemiological Data

A study by Waldram et al., analysing WGS of isolates of *Salmonella* belonging to the most commonly identified serotypes in England and Wales between April and August 2014. identified 32 clusters (566 of the 1445 isolates into 32 clusters of five or more) with a statistically significant epidemiological link for 17 clusters, whereas. during the same time frame, only one cluster was detected using the traditional methods [20]. The clusters were associated with foreign travel (n = 8), consumption of Chinese takeaways (n = 4), chicken eaten at home (n = 2), and one each of the following: eating out, contact with another case in the home and contact with reptiles [20]. This study provided a good evidence base that WGS can be used for the highly specific and highly sensitive detection of biologically related isolates prior to follow-up investigations to explore the epidemiological links. Waldram et al. concluded that clusters at the 0 and 5 SNP threshold should be prioritised for epidemiological investigation, as the meaningfulness of the relationship between isolates at the 10 SNP level was less clear. However, they noted that deeper epidemiological links may be uncovered by analysis at the 10 SNP level that may provide clues during an outbreak investigation [20].

A follow up study, by Mook et al., was undertaken in order to summarise how the WGS data was used to inform the implementation and development of a national gastrointestinal infection surveillance system [41]. This study retrospectively identified genetically related clusters of *S.* Enteritidis and *S.* Typhimurium infection during the first year of implementation (2014–2015) and determined the distribution of these clusters by the UKHSA operational levels [41]. A constrained WGS cluster definition based on the single nucleotide polymorphism distance (0,5,10 SNP SLC), case frequency and temporal spread was used. The study demonstrated that, in addition to the ability of WGS to identify small, geographically dispersed clusters, the resolution also enabled the identification of long-running, slowly developing clusters that may previously have remained undetected [41]. Previously, the detection of these types of clusters was difficult because the phenotypic methods, such as phage typing, did not provide a high resolution and they could only be performed on a selection of serovars, *S.* Typhimurium and *S.* Enteritidis, in which the majority had common phage types. Even when PFGE and MLVA could be used to provide further resolution, they were mainly used reactively to suspected *Salmonella* outbreaks, and not on a routine scale [8], whereas the resolution of the microbiological typing using WGS at the defined SNP levels clearly grouped strains into a microbiological cluster, irrespective of the time frame. We have now developed a greater understanding of the persistence of strains continuing in the community after the peak of an outbreak and becoming endemic in the population.

### 5.2. High Level Discriminatory Typing

It is well known that *S.* Enteritidis is the most common serovar in England and Wales for both sporadic and outbreak-related cases [8]. As described above, the previous methods, such as phage typing, were not as discriminatory as WGS and, further, subtyping methods, such as PFGE and MLVA, were used reactively where there was already an epidemiological signal of a potential outbreak. However, it is recognised that epidemiological investigations are often confounded by poor patient recall of the food they consumed before the onset of symptoms, which is challenging when trying to perform trace-back studies, particularly with the consumption of common food products such as chicken or eggs [34,37,43], and therefore may not give a strong statistical signal in an outbreak situation. This is further complicated if the source is in a product that is used as a base or sprinkled on salads [45] or due to cross-contamination [33] and therefore will not be easily identified in questionnaires. For example, the prospective use of routine WGS enabled the detection of three *S*. Enteritidis clusters that were further investigated [34,37,43] (Table 1, Appendix A). The investigation found these multi-country outbreaks to be associated with the consumptions of eggs imported into England from different countries [34,37,43] (Table 1, Appendix A). Eggs are a common food product and may not provide a strong epidemiological signal for investigation; without the use of high discriminatory WGS typing methods, these outbreaks could have continued to be under-detected.

### 5.3. Detection of Nested Clusters

In addition to WGS improving the discrimination between cases, it can also be used to understand the relatedness within a deeper genetic level with a phylogeny. For example, in June 2014, the *Salmonella* surveillance team were alerted to a hospital outbreak of *S*. Enteritidis 14b, a common PT associated with chicken eggs originating from outside of England [65,66,67]. The Epidemic Intelligence Information System (EPIS) alerted the UKHSA to six *Salmonella* outbreaks in France associated with eggs from a German producer and all of the strains had a single MLVA profile [34]. The analysis of the WGS data linked the outbreak to both a national and international egg distribution network, as well as nested clusters within a hospital and multiple transmission events linked to restaurants. These links were not detected with PT or MLVA profiling [34,35,40]. This fine resolution of the genomic data demonstrates our increased ability to detect nested clusters within large and ongoing outbreaks. In this example, the outbreaks relating to specific restaurants were nested in a national and international outbreak caused by the consumption of contaminated eggs [34,35,40] (Table 1, Appendix A). Other studies comparing PFGE to WGS have highlighted the considerable confidence that WGS affords in assigning an ‘indistinguishable’ status to two potentially linked bacteria [32,38,43].

### 5.4. Improved Case Ascertainment of Slow Burn Outbreaks

A retrospective analysis of the WGS data between 2012–2015 uncovered a previously undetected national outbreak of the common *S*. Enteritidis PT8 strain that had been on-going for four years [39]. A common food product could not be identified, but 30% of cases reported exposure to pet snakes. A robust case-definition based on the SNP profile and a case-control study around the exposure to reptiles or feed as risk factors showed that the exposure to snakes was the only variable independently associated with infection [39] (Table 1, Appendix A). This level of confidence in the microbiological typing data improved the case ascertainment during the outbreak investigations and further microbiological samples confirmed the feeder mice as the source of the contamination [39]. Mice destined to be fed to reptiles are not regarded as pet food and are not routinely tested for pathogenic bacteria. The study recommended the routine microbiological testing of feeder mice to ensure they are free from *Salmonella* contamination, again highlighting the importance of WGS microbiological methods for surveillance. The investigation provided an evidence base that the WGS forensic-level microbiological typing can be used to generate a robust case definition for case ascertainment and highlighted that the outbreak occurred under the surveillance radar for at least four years and was only detected with the adoption of routine WGS-led surveillance [39].

### 5.5. Evolutionary Context of Persistent Outbreaks

Phylogenomic based on the high-level resolution of the WGS data, as well as the utilisation of quantifiable genetic markers such as SNP typing, provides insight on the evolutionary context of outbreak strains. Analysis of the data from the *S.* Enteritidis PT14b dataset, held at the UKHSA, showed that by exploring the content of the deeper phylogenetic relationship between isolates, multiple transmission events had occurred in specific geographical regions in England and on a global scale. For example, a cluster of more than 30 cases of *S.* Enteritidis PT 14b, initially thought to be community acquired, was detected in a hospital in 2014. Phylogenetic analysis against the community background of strains showed two separate transmission events and confirmed there was an outbreak within the hospital [35]. Cases within the same 25 SNP cluster continued and were unrelated to the hospital; *Salmonella* enhanced surveillance, questionnaires and epidemiolocal analysis showed that there was a multi-country outbreak occurring (Table 1, Appendix A). In addition, there were multiple transmission events and multiple outbreaks, locally and nationally within England, which were traced back to the same German egg producer [34]. Using dated phylogeny generated by BEAST and egg supply network investigation, this further study showed a complex egg distribution network in the UK network consisting of several distinct distribution chains. They were able to show a clear statistical correlation between the topology of the UK egg distribution network and the phylogenetic network of the outbreak isolates [35]. These analyses showed that it is possible to map the vehicle and source of a complex and persistent outbreak using a phylogenetic approach, thus providing an evidence base to direct trace-back investigations to specific locations [34,35] (Table 1, Appendix A).

### 5.6. Inferring the Geographical Origin and/or Potential Animal Reservoir of a Foodborne Outbreak

Travel-related cases play a major role in Salmonellosis in England, as evidenced in Waldram et al.’s study, in which ~50% of the clusters detected by WGS were in association with travel [20]. Not all outbreaks have a strong signal with travel, but surveillance of travel is still integral when assessing clusters and can help to indicate the source of an outbreak. An example of this is the first UKHSA study, in which the prospective use of whole genome sequencing (WGS) detected a multi-country outbreak of *S.* Enteritidis in 2015 [37]. English cases were interviewed to obtain a food history and the links between suppliers were mapped to produce a food chain network for the chicken eggs and included the testing of food and environmental samples linked to the cases. Within the outbreak’s SNP defined cluster, 136 cases were identified in the UK and 18 in Spain. The association between the food chain network and the phylogeny was explored using a network comparison approach and a link was established between the UK, Spanish travel and Spanish cases [37]. This outbreak in England was linked with contemporaneous cases in Spain by WGS where 20% of cases had travelled to Spain. The study concluded that the UK and Spanish cases were exposed to a common source of *Salmonella*-contaminated chicken eggs [37] (Table 1, Appendix A). Other outbreaks may have a stronger travel signal with clusters; between 2014–2015 there was a multinational outbreak of travel-related *Salmonella* Chester infections in Europe [38]. Epidemiological and Microbiological outbreak investigations indicated that the cases were derived from food or from patients returning from Morocco, and that the cluster was likely a multi-source outbreak with several food vehicles contaminated by multidrug-resistant *S*. Chester strains [38].

### 5.7. Impact of WGS during the Investigation of Cross Border Outbreaks

The use of reporting and providing WGS data on multicounty platforms, such as the Epidemic Intelligence Information System (EPIS), now superseded by EpiPulse [68] and/or The Rapid Alert System for Food and Feed (RASFF) [69], enables cross-border outbreaks to be quickly identified in the same circulating strain or contaminated food product. In the instance where an outbreak, the detection of a clonally expanding strain or exposure to a food product poses an international risk, a joint risk outbreak assessment (ROA) is undertaken cross-border and globally published to alert stakeholders (Appendix A).

Food distribution is a global franchise; therefore, a large proportion of non-domestic outbreaks in England are due to imported food (Table 1, Appendix A). Prospectively detecting a foodborne outbreak, even at a local level, can subsequently reveal multi-country outbreaks. The combination of ECDC ROAs (Appendix A) and the digital nature of the WGS data allows the data to be readily exchanged and analysed between multiple institutions in different countries [35]. An example of this was the initial detection of *S*. Newport in a ready-to-eat watermelon slice from a supermarket retailer in England and communicated via RASFF [32]. Subsequent communications via EPIS and multi-country investigations found the same outbreak strain in six countries, linked to the consumption of watermelon originating from Brazil [32]. In some cases, an outbreak strain is first detected in other countries, such as the case with a multi-country outbreak of a novel *S*. enterica subspecies enterica serotype (11:z41:e,n,z15) associated with sesame seeds [45]. In response to an EPIS request, only a small number of cases were detected in England, which may have otherwise been assumed to be sporadic cases, and a direct link to a food product would have been difficult to trace. The advantage of global investigations is that the larger dataset and combined epidemiological analysis leads to a better understanding of the transmission rates and the sources of contamination. Outbreaks associated with food imports continue to occur, despite the strict EU regulations on the safety of imports of food related products [70], and with food imports opening up to other countries post-Brexit, we need to continue to be vigilant in checking the safety and enhance food testing.

## 6. The Limitations and Complexities of WGS Methods

### Defining Thresholds with a Salmonella Outbreak—One size Approach Does Not Fit All 

As described above, preliminary evaluation studies for the detection of clusters with respect to the epidemiological data have shown that 0–5 SNP level thresholds work well in detecting point source outbreaks and are more likely to have an obvious epidemiological link [20,41], as shown with eleven of the outbreaks highlighted in this study, covering a variety of food, travel and environmental exposures (Table 1). This will include the use of a five SNP threshold for point source outbreaks, such as foodborne contamination as described with watermelons [32] and mayonnaise [36], cross contamination environmental exposures such as drainage systems [33] or feeder mice [39], and even in the cases of more complex food chain networks, such as eggs [34,37,43]. For example, a re-evaluation of a 2014 multi-country European outbreak of *S*. Enteritidis phage type 14b associated with the consumption of eggs from a German producer [34], which formed a single five SNP single-linkage cluster with a maximum distance between any two genomes of 23 SNPs [40]. In this outbreak, with the analysis of the phylogeny, three clades could be separated within the monophyletic cluster, with a maximum difference of two SNPs, and all three clades could be traced to company X or its egg supply network, and the study postulated a common ancestor of all three clades which were genetically related at the 25 SNP level [40]. However, in the case of the 2015 multi-country European outbreak of *S*. Enteritidis, mentioned above, in relation to Spanish and English cases being exposed to a common source of *Salmonella*-contaminated chicken eggs [37], a ten SNP threshold was more appropriate. Hence, even in the cases of similar food exposures, the use of a SNP threshold needs to be dynamic and adapted according to the epidemiological investigation.

Bayesian approaches such as BEAST have proven useful for producing dated phylogeny to estimate the introduction of specific strains within a population and the measure of population size in case of intervention methods. However, these methods are greedy in terms of computing capacity, they are dependent on the size of the population to assess and request fine prior information (time sampling) and good sequencing quality, which makes them difficult to apply in real time. Due to the complexity of implementation, non-Bayesians methods, and specifically maximum likelihood phylogeny based on SNP differences, have been the implemented choice for GRBU to complemented clusters detected by the SNP address.

The use of employing single linkage SNP or allelic threshold for cluster detection for outbreak investigations helps to simplify the process, but it is important that we are dynamic with defining the threshold. Otherwise, this may over-simplify the definition and may lead to outbreaks being missed (false negatives) or clusters being defined as outbreaks when they are not (false positives), hence the importance of an epidemiological investigation. In some outbreaks, the use of SNP thresholds may not be appropriate at all, as was the case with a rare *S.* Adjame outbreak that had been detected in a local area in 2017 [44]. The sequencing of the *S.* Adjame outbreak strains produced a heterogeneous phylogeny showing multiple microbiological sub-clusters that were, in general, temporally and geographically linked, with some strains being genetically distinct (up to 1273 SNPs different). It was concluded that the contamination from an endemic source with a mixed population of strains could explain the heterogeneous genealogy, consistent with the hypothesized source of the contamination being imported herbs or spices from South Asia. This study again highlights the importance of integrated epidemiology when interpretating the WGS data and that WGS whole genome sequencing for outbreak detection and the use of SNP- or allele-based clustering methods as part of the case definition may not always be appropriate in heterogeneous outbreaks or rare/unusual serovars involving small numbers of cases [44].

## 7. Conclusions

This perspective provides an overview of the use of reference microbiology WGS data during food-borne outbreak investigations in England, demonstrating the key advantages, including: (i) establishing genomic approaches in the routine identification and detection of novel *Salmonella* lineages; (ii) robust comparative SNP and cgMLST methodologies, enabling international collaboration; (iii) an unprecedented level of strain discrimination, including the detection of nested clusters; (iv) improved case ascertainment; (v) an insight into the evolutionary context for emerging pathogenic strains; and (vi) inferring the geographical origins from the phylogenetic signal. Collecting epidemiological data is essential when interpretating phylogenetic clusters, and defining clusters by the number of SNP or allelic differences between isolates provides information regarding the strain relatedness. Isolates originating from the same source population with less time to diverge form a common ancestor and are more likely to have fewer nucleotide or allelic differences between a pair of isolates. However, when analysing a source population, the amount of diversity is dependent on the duration of the infection, the effective size of the population and the time colonised in a particular niche. Hence, the definition of the absolute thresholds of nucleotide or allelic differences for exclusion and inclusion within an outbreak can vary, and the epidemiological information should always be used, where possible, to inform the outbreak definition. With the expansion of WGS-based typing analysis on a global scale, we will continue to see an improvement of trace-back investigations during food-borne outbreaks. This will ensure the rapid implementation of interventions to protect public health. The utility of the publicly available WGS databases linked to the clinical, epidemiological and environmental context of each strain cannot be underestimated for the purpose of the risk assessment and management of food-borne outbreaks. This review provides an evidence base that the genomic methods at the GBRU continue to be a fit for purpose test, not just for the initial report of typing data, but for further downstream analysis in a public health setting.

## Figures and Tables

**Figure 1 pathogens-12-00223-f001:**
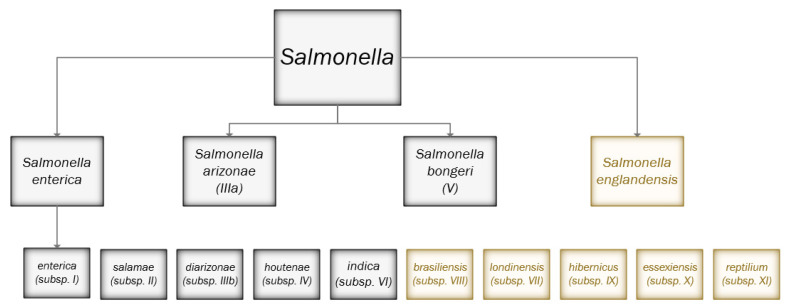
Genomic Identification of *Salmonella* species and subspecies. Overview of the taxonomy of *Salmonella* currently recognised (in grey boxes) and proposed (in yellow) in the post-genomic era. Pearce et el describes novel subspecies *S. enterica* subspecies *londinensis* (VII), subspecies *brasiliensis* (VIII), subspecies *hibernicus* (IX) and subspecies *essexiensis* (X) and an additional novel subspecies, *reptilium* (XI) Further, genomic analyses indicated that *S. enterica* subspecies *arizonae* (IIIa) isolates is divergent from the other *S. enterica* subspecies and is sufficiently distinct to be classified as a separate species, *S. arizonae* [62]. Since the Pearce et al. study, another novel species *S. englandensis* has also been detected (Chattaway, unpublished).

**Table 1 pathogens-12-00223-t001:** Overview of published Salmonella Outbreaks in England assessed by whole genome sequencing, 2012–February 2020.

Published Year (Reference)	Description of Study (eBG, ST, SNP Address Where Available) #No. of Cases	Typing Methods	SNP Threshold Detection ^†^ Clustering Threshold	UK-National	Nternational	Domestic Source	Non-Domestic Source	Association with Travel	Foodborne	Person to Person Transmission	Environmental Contamination	Short-Term (>6 Months)	Medium-Term (6–12 Months)	Long-Term (>12 Months)	Possible Source (Not All Statistically Confirmed)	Benefits of the WGS Approach and Context
2012 [32]	A Multi-Country Salmonella Newport outbreak associated with watermelon imported from Brazil, UK, Germany, October 2011-January 2012 #63 cases	**WGS** SNP	5		√		√		√					√	Watermelon	Robust, high level strain discrimination compared with traditional molecular typing methods. Inferring the geographical origin of an outbreak strain from the phylogeny at the national and international level. Non-Domestic source of outbreak strains. Importance of one health approach routine surveillance of food typing
2015 [33]	A prolonged restaurant outbreak of Salmonella Typhimurium linked to the building drainage system, February 2015-March 2016 (eBG1, ST34, 1.1.1.124.395.395.x) #82 cases	PT MLVA **WGS** SNP	5	√		√					√			√	Structural faults with the drainage system and ineffective drain water-traps	Robust, high level strain discrimination compared with traditional molecular typing methods. Inferring the geographical origin of an outbreak strain from the phylogeny at the national level. Evolutionary context of outbreak strains Forensic level typing for case ascertainment. Domestic source of outbreak strains.
2015 [34]	A multi-country Salmonella Enteritidis phage type 14b outbreak associated with eggs from a German producer, United Kingdom, May to September 2014. (* 1.2.3.38.38.38.x, current address 1.2.3.18.38.38) #287 cases	PT MLVA **WGS** SNP	5		√	√	√		√	√	√	√			Chicken eggs from a German Producer	Robust, high level strain discrimination compared with traditional molecular typing methods. Inferring the geographical origin of an outbreak strain from the phylogeny at the national level and international level. Forensic level typing for case ascertainment. Differentiation of multiple transmission events including separate transmission outbreak within a hospital. Evolutionary context of outbreak strains. Domestic and Non-Domestic source of outbreak strains.
2016 [35]	Multiple Salmonella Enteritidis phage type 14b outbreaks associated with eggs from a German producer, United Kingdom, May to September 2014. Phylogenetic analysis of the network of the implicated foodstuff (* 1.2.3.38.38.38.x, current address 1.2.3.18.38.38) #350 cases	PT MLVA **WGS** SNP	5		√	√	√		√	√	√	√			Five point-source outbreaks associated with three Chinese restaurants, a hospital and kebab grill all traced-back to one specific German egg producer	Robust, high level strain discrimination compared with traditional molecular typing methods. Inferring the geographical origin of an outbreak strain from the phylogeny at the national level and international level. Forensic level typing for case ascertainment. Differentiation of five multiple transmission events including multiple restaurants and an outbreak within a hospital. Evolutionary context of outbreak strains. Domestic and Non-Domestic source of outbreak strains.
2015 [36]	A local outbreak of Salmonella Typhimurium DT8 linked with mayonnaise using raw eggs, Jersey, July–August 2013. #21 cases	PT **WGS** SNP	5, 50	√		√			√			√			Mayonnaise made using raw eggs	Robust, high level strain discrimination compared with traditional molecular typing methods. Inferring the geographical origin of an outbreak strain from the phylogeny at the national level. Evolutionary context of outbreak strains. Domestic source of outbreak strains.
2017 [37]	A multi-country Salmonella Enteritidis of multiple phage types outbreak associated with contaminated eggs, UK, Spain, February–October 2015 (eBG4, ST11, 1.2.3.151.362.x) #154 cases	PT **WGS** SNP	10		√		√	√	√				√		UK and Spanish cases exposed to a common source of contaminated chicken eggs.	Robust, high level strain discrimination compared with traditional molecular typing methods. Inferring the geographical origin of an outbreak strain from the phylogeny at the national level and international level. Forensic level typing for case ascertainment. Evolutionary context of outbreak strains. Non-Domestic source of outbreak strains.
2017 [38]	Multinational outbreak of Morocco travel-related Salmonella Chester infections in Europe, strains ranging between 2011–2015. (eBG49, ST1954, 1.1.1.1.x) #162 cases	MLVA PFGE **WGS** SNP	25				√	√	√	√			√		Shrimp and Chicken Sausage, travel to Morocco	Robust, high level strain discrimination compared with traditional molecular typing methods. Inferring the geographical origin of an outbreak strain from the phylogeny at the international level. Forensic level typing for case ascertainment. Differentiation of possible separate transmissions of food source associated with antimicrobial resistance markers. Evolutionary context of outbreak strains. Non-Domestic source of outbreak strains.
2017 [39]	National outbreak of Salmonella Enteritidis associated with imported reptile feeder mice in the United Kingdom (eBG4, ST11, 1.5.159.280.280.280.x) #147 cases	**WGS** SNP	5	√			√				√			√	Reptile feeder mice	Analysis of whole genome sequencing data uncovered a previously undetected outbreak of *Salmonella* Enteritidis that had been on-going for four years. Inferring the geographical origin of an outbreak strain from the phylogeny at the international level. Forensic level typing for case ascertainment. Evolutionary context of outbreak strains. Non-Domestic source of outbreak strains.
2017 [40]	Multiple Salmonella Enteritidis phage type 14b outbreaks associated with eggs from a German producer, Europe, March to November 2014. Phylogenetic analysis of the network of the implicated foodstuff (* 1.2.3.38.38.38.x, current address 1.2.3.18.38.38) #400 cases	PT **WGS** SNP	5		√	√	√		√	√	√		√		Multiple point-source outbreaks and sporadic cases associated with multiple vehicles) traced-back to one specific German egg producer	Robust, high level strain discrimination compared with traditional molecular typing methods. Inferring the geographical origin of an outbreak strain from the phylogeny at the national level and international level. Forensic level typing for case ascertainment. Differentiation of multiple transmission events in multiple countries and vehicles. Evolutionary context of outbreak strains. Domestic and Non-Domestic source of outbreak strains.
2018 [41]	Retrospectively identifying genetically related clusters of Salmonella Enteritidis and Typhimurium over a 1-year period using differing SNP detection thresholds and determining the distribution of these clusters by UKHSA operational levels.	**WGS** SNP	0, 5, 10	√	√	√	√	√	√	√	√	√	√	√	Multiple sources	Robust, high level strain discrimination compared with traditional molecular typing methods. Forensic level typing for case ascertainment. Differentiation of multiple outbreaks, locally, regionally and nationally. Domestic and Non-Domestic source of outbreak strains.
2018 [20]	Retrospectively identifying genetically related clusters of common Salmonella serotypes over a 1-year period using differing SNP detection thresholds	**WGS** SNP	0, 5, 10	√	√	√	√	√	√	√	√	√	√	√	Multiple sources	Robust, high level strain discrimination compared with traditional molecular typing methods. Forensic level typing for case ascertainment. Differentiation of multiple outbreaks, locally, regionally and nationally. Domestic and Non-Domestic source of outbreak strains.
2018 [42]	Evaluation of a core genome multilocus typing (cgMLST) scheme for the high-resolution reproducible typing of Salmonella enterica isolates, by its application to a large European outbreak of S. Enteritidis. (* 1.2.3.38.38.38.x, current address 1.2.3.18.38.38) #350 cases	PT **WGS** SNP cgMLST	5		√	√	√		√	√	√	√			Chicken eggs from a German Producer	Robust, high level strain discrimination compared with traditional molecular typing methods. Inferring the geographical origin of an outbreak strain from the phylogeny at the national level and international level. Forensic level typing for case ascertainment. Differentiation of multiple transmission events. Identification of relationships between the date of isolation and the spread of the cgMLST types. Evolutionary context of outbreak strains. Domestic and Non-Domestic source of outbreak strains.
2019 [8]	Identification of genetically related clusters of all Salmonella eBG over a 2-year period and determining characteristics such as size and age of cluster.	**WGS** SNP	5	√	√	√	√	√	√	√	√	√	√	√	Multiple sources	Robust, high level strain discrimination compared with traditional molecular typing methods. Detection of all *Salmonella* clusters over a two-year period. Description of the characteristics of *Salmonella* whole sequencing genome clusters. Assessment of cluster burden by eBG
2019 [43]	A large multi-country outbreak of Salmonella enterica serotype Enteritidis in the EU and European Economic Area (* 1.2.3.175.175.175.x, current address 1.2.3.18.175.175.x) #838 confirmed cases #371 probable cases	MLVA PFGE **WGS** SNP	5		√		√	√	√		√			√	Chicken eggs from a Polish Producer	Robust, high level strain discrimination compared with traditional molecular typing methods. Inferring the geographical origin of an outbreak strain from the phylogeny at the national level and international level. Forensic level typing for case ascertainment. Differentiation of multiple transmission events. Evolutionary context of outbreak strains Non-Domestic source of outbreak strains.
2019 [44]	An atypical local outbreak of Salmonella Adjame investigated using SNP and cgMLST analysis, London, May to July 2017. (eBG421, ST3929, 1.1.1.1.1.1.1—Cluster 1; (eBG421, ST3929, 1.8.8.8.8.8.8—Cluster 2; eBG421, ST4023, 1.2.2.2.2.x—Cluster 3; Other variable SNP addresses) #14 cases	**WGS** SNP cgMLST	N/A	√			√		√			√			Herbs and spices from local South Asian speciality grocery stores.	Robust, high level strain discrimination compared with traditional molecular typing methods. Cases linked in time and place but WGS showed marked heterogeneity, atypical of a point source *Salmonella* outbreak. Differentiation of multiple clusters in association with sample date including a previously undetected cluster. Highlights the complexity of case ascertainment and that specific genomic types may not always be appropriate. Non-Domestic source of outbreak strains.
2019 [45]	A Multi-Country outbreak with a novel Salmonella enterica subspecies enterica serotype (11:z41:e,n,z15) associated with Tahini, Europe, March 2016 to April 2017 (ST2914—1.1.1.1.1.1.x) #47 cases	**WGS** SNP	5 (cgMLST)		√		√		√					√	Tahini, a paste made from hulled, ground and toasted sesame seeds from Greece	Robust, high level strain discrimination compared with traditional molecular typing methods. Inferring the geographical origin of an outbreak strain from the phylogeny at the international level. Forensic level typing for case ascertainment. Differentiation of multiple transmission events. Non-Domestic source of outbreak strains.
2020 [46]	Evaluation of multiple SNP and cgMLST scheme for the high-resolution reproducible typing of Salmonella enterica isolates, by its application to a large European outbreak of S. Enteritidis. (* 1.2.3.175.175.175.x, current address 1.2.3.18.175.175.x) #180 cases	MLVA **WGS** SNP cgMLST	5		√		√		√					√	Chicken eggs from a Polish Producer	Robust, high level strain discrimination compared with traditional molecular typing methods. Evolutionary context of outbreak strains. Distinction of two separate clusters. High concordance of six distinct workflows, in use by several European institutions to identify concordant clusters of genetically related *Salmonella* Enteritidis; thus, allowing identification of cross-border outbreaks. The use of an unsupervised machine learning methodology to detect an optimal number of clusters that separate outbreak from non-outbreak isolates. Non-Domestic source of outbreak strains.

* The SNP address has since changed due to a merge and the current SNP address is also shown, please see Appendix A for further detail. ^†^ The SNP thereshold detection indicates at which level the outbreak was first detected. Definitions—National—occurring within the UK, international—occurring outside the UK and in multiple other countries, Domestic source–source is from within the UK, Non-Domestic source—source is outside of the UK including imported food. Short-, medium- and long-term categories refer to the duration of the outbreak in which months are stated in the column header. The SNP linkage clustering threshold is the level at which the cluster was assessed as part of the analysis and relates to any two strains linked within the threshold specified within a defined cluster. The strain distance will vary across the cluster depending on the size of the cluster. # Number of cases stated are at time of publication of the specified reference.

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
