# Peer review of "Evaluation of Genomic Typing Methods in the Salmonella Reference Laboratory in Public Health, England, 2012–2020"

_pathogens, 2023, doi:10.3390/pathogens12020223_

Round 1

Reviewer 1 Report

In this review, Chattaway and colleagues aim to understand if the current methods used in the reference laboratories in England are still suitable for typification, epidemiology monitoring and outbreak tracing according to the current norms and accreditation procedures.

This review is relevant at the public reference laboratory and offers an interesting perspective for evaluating current protocols used for this unit to inform and manage public health risks while fulfilling the ISO 15189 standard regulations of continuous assessment. I have only a few suggestions that may strengthen the quality of the manuscript.

1.- This paper could benefit from a short discussion regarding the global application of WGS typing methods used in other reference laboratories worldwide and their results, as this could help build a statement of the broader application of these techniques in public health globally.

2.- To show the options of the current and newer methods available and their possible advantages or disadvantages in large-scale implementation, the authors could mention the existence of other WGS subtyping methods that could fit the porpoises of evolutionary analysis, epidemiology monitoring and outbreak traceability, such as CRISPR typing (PMID: 29212058), prophage profile (PMID: 29780368), rMLST and wgMLST (PMID: 29621240).

3.- The clustering methods in SNP typing could be more problematic and could mislead the fast-tracking of an outbreak due to the different implementations across laboratories. Recently different methods have been developed, such as the algorithm named rPinecone (PMID: 30920366), which shows some advantages in detecting lineages in clonal populations. It could be interesting if the authors discuss the implementation of this type of tools in light of their current methods and if the new algorithms could improve the reproducibility across reference laboratories.

4.- It would also complement the manuscript to discuss in the section named “The limitations and complexities of WGS methods” the possible difficulties of the implementation of retrospective Bayesian approaches using BEAST (mentioned in lines 381–387) for tracing back outbreaks and explore whether non-Bayesian methods could be more suitable for reference laboratories.

Minor comments

5.- Supplementary Table 1 is titled Table 2.

Author Response

One   Thank you for the feedback
  1.- This paper could benefit from a short discussion regarding the global application of WGS typing methods used in other reference laboratories worldwide and their results, as this could help build a statement of the broader application of these techniques in public health globally. Sentence added at end of section 4.3 in regards to  reference laboratories using SNP and cgMLST and using different methodologies for inter laboratory comparison analysis, which has been useful in global outbreak investigations as evidenced in published papers and ROA which i have referenced to.  Line 300
  2.- To show the options of the current and newer methods available and their possible advantages or disadvantages in large-scale implementation, the authors could mention the existence of other WGS subtyping methods that could fit the porpoises of evolutionary analysis, epidemiology monitoring and outbreak traceability, such as CRISPR typing (PMID: 29212058), prophage profile (PMID: 29780368), rMLST and wgMLST (PMID: 29621240) Thank you for the excellent suggestion, a comparison analysis of different methodologies would be beneficial to the scientific community and would be a worthwhile paper in itself. Unfortunately, this is outside the scope of this paper as we were analysing WGS methods that we are currently using in our laboratory and don't have the resources to compare our outbreaks to other methods.
  3.- The clustering methods in SNP typing could be more problematic and could mislead the fast-tracking of an outbreak due to the different implementations across laboratories. Recently different methods have been developed, such as the algorithm named rPinecone (PMID: 30920366), which shows some advantages in detecting lineages in clonal populations. It could be interesting if the authors discuss the implementation of this type of tools in light of their current methods and if the new algorithms could improve the reproducibility across reference laboratories. Thank you for highlighting other algorithms available for cluster assessment, comparing against other algorithms would be a useful study as a separate analysis. Unfortunately, this is outside the scope of this paper as we were analysing WGS methods that we are currently using in our laboratory and don't have the resources to compare our outbreaks to other algorithms.
  4.- It would also complement the manuscript to discuss in the section named “The limitations and complexities of WGS methods” the possible difficulties of the implementation of retrospective Bayesian approaches using BEAST (mentioned in lines 381–387) for tracing back outbreaks and explore whether non-Bayesian methods could be more suitable for reference laboratories. Thank you for this comment. We have added comments on the difficulties and  clarified that we are not routinely using bayesian approaches, but maximum likelihood phylogeny to corroborated clusters detected by SLC. Line 494

Reviewer 2 Report

Nice paper that is of international significance in its breadth and importance.  Hard work is all done, but table 1 is not as good as it could be.

Table 1 should stand alone without having to dig too far into the paper. I think the case numbers from the supplementary table 1 should go in the main paper. These are BIG outbreaks.  The tick boxes I think could be trimmed a little and certainly need a legend to define better. What is difference between UK-National and Domestic?  Short term, medium term long term need defining. Maybe better as time scale which could combine expressing in months.  What does association mean?    Published Year is not correct if referring to publication.  And not really needed as have dates in Description where could add the reference and do away with column of published year. Papers covering multiple sources how many actual source identified - worthwhile including examples of sources, or splitting into identified sources, and clusters with unknown source if that is the case.

The SNP linkage column needs definition, and with these very large clusters I think it would be helpful to include the SNP range between any two isolates. Eg reference 46, in table simple 5 SNP linkage, but as highlighted in text difference between any two isolates is a maximum of 25 SNPs. If you had a less complete data set (ie only isolates at each end) they wouldn't be in 5 SNP cluster. Think this paper could mislead people as to the level of differences that might observe.

Line 450/451. SNP already been defined, so just need SNP

Author Response

Two Nice paper that is of international significance in its breadth and importance.  Hard work is all done, but table 1 is not as good as it could be. Thank you for the feedback
  Table 1 should stand alone without having to dig too far into the paper. I think the case numbers from the supplementary table 1 should go in the main paper. These are BIG outbreaks.  The tick boxes I think could be trimmed a little and certainly need a legend to define better. What is difference between UK-National and Domestic?  Short term, medium term long term need defining. Maybe better as time scale which could combine expressing in months.  What does association mean?    Published Year is not correct if referring to publication.  And not really needed as have dates in Description where could add the reference and do away with column of published year. Papers covering multiple sources how many actual source identified - worthwhile including examples of sources, or splitting into identified sources, and clusters with unknown source if that is the case. Table 1 has been updated, national, international, domestic and non-domestic, Short, medium and long term categories  definitions added as a foot note, published year removed. Colums updated to state national and internation for consistent terminology.

No. of cases has been added and related to the published paper, this has been added as a foot note for clarity.

Possible source is specified as a column and states the source idenfied in the outbreak investigation, epidemiological investigations do not always have a clear statistically confirmed signal, hence why the word associated is stated. It is complex to summarise these detaisl in a table and therfore readers can delve into the referenced paper for further detail if required in relation to statistical methods.

Tried to reduce the Tick boxes  but these relate to specific distinct categories and therefore could not be cut down.
  The SNP linkage column needs definition, and with these very large clusters I think it would be helpful to include the SNP range between any two isolates. Eg reference 46, in table simple 5 SNP linkage, but as highlighted in text difference between any two isolates is a maximum of 25 SNPs. If you had a less complete data set (ie only isolates at each end) they wouldn't be in 5 SNP cluster. Think this paper could mislead people as to the level of differences that might observe. Thank you for the comments. We have amended the column title and defined in the footnote.

In the methodology and discussions we have described that since we used single linkage clustering and some outbreak are long term persisiting and/or large, the clustering will allowed isolates to accrued SNP mutation but still be labelled as part of same cluster. Phylogeny is used to make sense of theses 5SNPs group where maximum distance between isolates can be greated that 5SNPs. In this study, we have focused on the Salmonella reference laboratory where we routinely sequence all our isolates and therefore the dataset is complete.

In addition, we have updated the technical supplementary information sheet to explain how the SNP typing works and clarify that the maximum SNP distance can vary as the cluster expands.
  Line 450/451. SNP already been defined, so just need SNP corrected

Reviewer 3 Report

The current study reviews the Salmonella typing and the utility of WGS for the detection and investigation of outbreaks. This kind of study is helpful for researchers to share the current situation of microbes and their genetics in different countries around the world. However, the authors have to improve the quality of their data presentation.

-  Please check the typos in the whole manuscript, for instance, lipopolysaccharide (0) antigens are O-antigens, etc...

- The present study could be helpful for physicians and public health coworkers, please simply describe what is the WGS and its purpose in the introduction.

- The most readable sections in this study are the introduction and the conclusion, so please improve the introduction. What is the clinical importance of Salmonella and why it is needed to trace its species and the changes in their genomes? .... etc.

Author Response

Three The current study reviews the Salmonella typing and the utility of WGS for the detection and investigation of outbreaks. This kind of study is helpful for researchers to share the current situation of microbes and their genetics in different countries around the world. However, the authors have to improve the quality of their data presentation. Thank you for the feedback
  -  Please check the typos in the whole manuscript, for instance, lipopolysaccharide (0) antigens are O-antigens, etc... corrected
  - The present study could be helpful for physicians and public health coworkers, please simply describe what is the WGS and its purpose in the introduction. Introduction expanded to include  a background to the clinical and public health importance of Salmonella and the applications of WGS. Line 33-52 and line 66-71
  - The most readable sections in this study are the introduction and the conclusion, so please improve the introduction. What is the clinical importance of Salmonella and why it is needed to trace its species and the changes in their genomes? .... etc. Introduction expanded to include  a background to the clinical and public health importance of Salmonella and the applications of WGS. Line 33-52 and line 66-71

Round 2

Reviewer 1 Report

This study offers bases that may have a broader influence to improve the epidemiological surveillance of Salmonella. I have no further comments.

Reviewer 3 Report

The raised point are well addressed and I think it can be published now.